# Evaluation of the Uniformity of Protective Coatings on Concrete Structure Surfaces Based on Cluster Analysis

**DOI:** 10.3390/s21165652

**Published:** 2021-08-22

**Authors:** Dunwen Liu, Wanmao Zhang, Yu Tang, Yinghua Jian, Chun Gong, Fengkai Qiu

**Affiliations:** School of Resources and Safety Engineering, Central South University, Changsha 410083, China; dunwen@csu.edu.cn (D.L.); 205512136@csu.edu.cn (W.Z.); jyh__0412@csu.edu.cn (Y.J.); gongchun@csu.edu.cn (C.G.); 165511052@csu.edu.cn (F.Q.)

**Keywords:** concrete protection, infrared detection, image processing, cluster analysis, uniformity evaluation

## Abstract

With the continuous development of urbanization and industrialization in the world, concrete is widely used in various engineering constructions as an engineering material. However, the consequent problem of durability of concrete structures is also becoming increasingly prominent. As an important additional measure, a protective coating can effectively improve the durability of concrete performance. Moreover, the uniformity of the concrete surface coating will directly affect its protective effect. Therefore, we propose a nondestructive inspection and evaluation method of coating uniformity based on infrared imaging and cluster analysis for concrete surface coating uniformity detection and evaluation. Based on the obtained infrared images, a series of processing and analysis of the images were carried out using MATLAB software to obtain the characteristics of the infrared images of the concrete surface. Finally, by extracting the temperature distribution data of the pixel points on the concrete surface, an evaluation method of concrete surface coating uniformity based on a combination of cluster analysis and hierarchical analysis was established. The evaluation results show that the determination results obtained by this method are consistent with the actual situation. This study has a positive contribution to the testing of concrete surface coating uniformity and its evaluation.

## 1. Introduction

Concrete structures are subjected to chloride ion attack, sulfate corrosion, and carbonation corrosion during service. These accelerate the deterioration of material properties and can lead to structural failure or even destruction in severe cases [1]. In the early 20th century, statistical studies on concrete corrosion found that the cost of repair and reinforcement of structures due to concrete corrosion was several times higher than the cost of new construction. The results of corrosion cost surveys in China found that corrosion costs in 2014 were as high as RMB 2.1 trillion, accounting for 3.34% of the GDP in that year [2]. Therefore, the corrosion of concrete structures not only involves structural safety but also involves ecological civilization issues, energy conservation issues, and economic issues.

Experts from different countries have carried out a lot of research for concrete corrosion protection. Concrete material ratios and microstructures have been optimized. These not only enhance the strength of concrete but also improve the anti-corrosion performance [3]. However, with the development of civil engineering structures towards deep sea and mountainous areas, coupled with the frequent occurrence of global climate extremes, the structured environment has become more complex. The structure is not only subject to dynamic and static loading, and the corrosion of the structure by environmental and biological factors cannot be ignored. Therefore, in order to improve the corrosion resistance and durability of concrete structures, protection of the surface of critical parts of concrete structures is proposed [4]. Under the premise of not changing the property of concrete material, the protective coating can increase the functional characteristics of concrete. Moreover, it is widely used in engineering construction. Liliana Baltazar [5], Sang-SoonPark [6], and others have conducted extensive studies on the enhancement effect of inorganic silicate-type coatings such as sodium silicate on concrete surfaces. In contrast to inorganic silicate coatings, research on organic and hybrid coatings has focused more on the development and preparation of new coatings. T.S. Velayutham [7], Aruz Petcherdchoo [8], and Paola Scarfato [9] prepared polyurethane (PUR) coatings, organosilane coatings, and polymer clay nanocomposite coatings. composite coatings, etc.

However, the construction process of spraying protective coatings on the surface of concrete structures is prone to defects such as uneven thickness, porosity, and inclusions [10]. These defects can affect the protective performance of the coating to varying degrees and can even lead to coating protection failure—especially when the anticorrosive silane spraying on the surface of the concrete pipe sheet in the tunnel is not uniform. The aggressive ions contained in the water body can easily pass through the weak part of the silane spraying on the surface of the pipe sheet, and cause local corrosion to the pipe sheet [11]. Therefore, the uniformity test of silane spraying on tunnel pipe sheets is extremely important for its corrosion resistance and durability during operation.

At present, there are few studies on the uniformity testing of spraying protective coatings on concrete surfaces. In the actual engineering application, this mainly relies on the visual inspection method. The method is a mostly subjective judgment by the naked eye of the inspector, which lacks a scientific basis and its accuracy is difficult to be guaranteed. In recent years, with the improvement of the accuracy of infrared imaging systems and the rapid development of image processing technology, infrared thermal imaging technology has gradually made significant breakthroughs in modern industry, medical and biological fields [12,13,14]. Infrared imaging, as a nondestructive testing and analysis tool, has demonstrated powerful advantages. The study of its detection mechanism and application fields has received a lot of attention from scholars. The current research content on infrared imaging detection technology mainly focuses on the design of infrared imager systems, applied excitation method, and infrared image processing and applicability [15,16,17]. In the application of infrared imaging inspection technology, Xu Hongguo [18] used infrared imaging and temperature sensor monitoring equipment to detect concrete defects and established a nondestructive concrete inspection method based on infrared thermography. Ying Xu [19] proposed an optical excitation line laser thermal source infrared thermography method for debonding detection of FRP reinforced concrete structures for detecting debonding of FRP reinforced concrete. Ptacek Lisa [20] introduced a nondestructive inspection method for concrete curing quality by near-infrared hyperspectral imaging, and the results showed that the method was highly reliable for distinguishing different curing types of concrete. Lu Yang [21] used infrared thermography to take thermal images from the surface to assess the effect of subsurface defects on the sensitivity and accuracy of detection. Jang Keunyoung [22] proposed an autonomous detection technique for concrete cracks based on deep learning by combining hybrid images of visual and infrared thermal imaging images. Therefore, infrared thermal imaging technology has a broad application prospect in the analysis of uniformity detection of protective coatings on concrete surfaces.

In general, the temperature resolution and contrast of the images obtained from thermal imaging cameras are generally low due to the limitations of their own performance. Moreover, the images also contain various noises, which bring a very negative impact on the detection and analysis of the target. Based on these problems, many researchers abroad have conducted in-depth studies on the correction of detector inhomogeneities, and the removal of periodic noise, etc. [23,24,25,26]. The research results have an important role in promoting the development of infrared image processing technology. However, there are still certain errors in the processed infrared images. In order to realize the use of infrared imaging technology to detect the uniformity of concrete surface coating, it is necessary to carry out a series of processing of infrared images. In this paper, based on the MATLAB software function tool, the infrared thermal images obtained from the experiments are processed in a relevant way. Finally, the uniformity of the coating is determined qualitatively by the distribution of the surface areas of the images.

In the process of infrared imaging inspection experiments, the thermal imaging camera collects the surface temperature distribution of each concrete specimen at different frames. For the massive amount of data, it is impossible to continue the analysis by the traditional manual analysis means. With the rapid development of modern computer operation speed and the in-depth research on data processing methods, it is possible to use the artificial intelligence method to confirm the results of experimental data prediction with those obtained from engineering experiments [27]. Clustering analysis is an unattended learning method in machine learning in the field of artificial intelligence. In addition to machine learning, it can be used for statistics, spatial data mining, and image recognition. In this statistical algorithm, grouping programs such as S-Plus, SPSS, and SAS are used intensively, which utilize K-means and cluster analysis methods [28]. There are two types of clustering in MATLAB, including hierarchical clustering and K-means clustering [29,30]. K-means clustering was invented by MacQueen in 1967 [31] and it is one of the most commonly used unattended learning methods. K-means clustering possesses an assignment mechanism that allows each dataset to belong to only one cluster so that each point in the dataset is assigned to its nearest node clustering [32,33]. The ease of implementation and fast operation on large datasets are the main advantages of K-means clustering. Menesatti Paolo [34], Shoa Pedram [35], and Yousefi Bardia [36] et al. successfully applied the clustering analysis method to cluster infrared thermal images. These studies provide important references for the evaluation of the homogeneity of silane coatings.

Therefore, this study addresses the current problems of homogeneity testing of silane coatings. Infrared imaging non-destructive testing technology is used, combined with cluster analysis to statistically analyze the infrared imaging data, so as to achieve the evaluation of the homogeneity of silane coatings. The main research objectives include: (1) carry out experimental research on infrared imaging inspection of concrete surface coatings and obtain the temperature distribution images of concrete coating structure surfaces; (2) perform a series of processing of the acquired infrared images based on the MATLAB software and qualitatively determine the homogeneity of the concrete surface coating using morphological processing methods; (3) according to the large sample data of temperature distribution of each pixel point on the concrete surface at different moments, a new method based on a combination of cluster analysis and hierarchical analysis is proposed for determining the uniformity of concrete surface coating.

## 2. Materials and Methods

### 2.1. Infrared Imaging Detection of Silane Coatings

When a protective coating is applied to the surface of a concrete specimen, the protective coating forms a coating structure on the surface. If the sprayed protective coating is uniform on the surface of the specimen, the changes in density, thermal conductivity, and specific heat capacity of each part of the coating structure should be the same. According to the basic theory of heat transfer, the distribution of temperature change characteristics caused by the coating on the surface is also consistent. If the protective coating on the surface of the specimen is not uniform, there will be variability in the distribution of the temperature change characteristics. Thus, the degree of uniformity of the protective coating on the surface of the concrete structure can be evaluated qualitatively based on these differences.

#### 2.1.1. Infrared Imaging Test Principle

An infrared imaging system is essentially a device that converts different electrical signals based on the temperature and emissivity of the natural scene. It generates a displayable infrared image by mapping the received infrared radiation energy of a target into grayscale values through an infrared detector. When the intensity of the infrared radiation of the target is higher, its grayscale value reflected in the infrared image is larger [37]. The infrared imaging system mainly consists of an electrical signal processing system, an image processing system, an optical system, and an infrared detector. The working principle of the IR imaging system is as follows.

(1) Infrared radiation from the target, background, and various interferences is transmitted in the atmosphere and reaches the infrared optical system after a certain attenuation effect.

(2) The optical system focuses the infrared radiation and then transmits the radiation to the infrared detector.

(3) Infrared detectors convert the received infrared radiation into current signals of different strengths and weaknesses.

(4) Transmission of the output electrical signal to an electrical signal processing system and enhancement by an amplifier therein.

(5) The amplified electrical signal is transformed into a two-dimensional graphic that can be viewed directly on a computer after passing through an image processing system.

#### 2.1.2. Infrared Imaging Test Program

The main component of this test silane coating is 2,4,4-trimethylpentyltriethoxysilane with molecular formula C_14_H_32_O_3_Si. Silane is a colorless liquid at room temperature, and its boiling point is 236 °C. The test used an infrared bulb as a thermal excitation source to apply continuous heat flow excitation to the concrete structure. In addition, an infrared imaging system was constructed with an SC7000 infrared camera from FLIR.

Considering the properties of the concrete specimen itself can affect the test results, the specimen was obtained by coring from the same subway tunnel pipe piece. The specimens were made into cylinders of 5 cm in diameter and 10 cm in height. Three groups of tests were designed according to the different degrees of uniformity of the sprayed silane. They were classified as more uniform (N1), non-uniform (N2), and very non-uniform (N3). The infrared detection test system is shown in Figure 1, and the flow chart of this study is shown in Figure 2. The specific steps of the test are as follows.

(1) Placement of the thermal imaging camera

Use the triangle bracket to place the thermal imaging camera in a suitable position. The infrared thermal imager is adjusted to the horizontal center position so that the lens of the thermal imager and the test piece are at the same horizontal height.

(2) Placement of concrete specimens

Use the bracket to fix the two infrared bulbs and adjust the corresponding position. Take a concrete specimen A_1_ and place it in the uniform temperature field formed by the two infrared bulbs to ensure the uniform heating of the specimen surface. The distance between the lens of the thermal imager and the test piece is about 1 m.

(3) Connect infrared equipment and set parameters

Connect the computer interface to the GigE vision port of the infrared camera with an Ethernet cable, and connect the data collected by the camera to the infrared image processing software through the CNUC management tool in the computer.

(4) Preheat the infrared bulb and camera

After connecting the power supply, turn on the switch of the infrared bulb and the power supply of the camera for preheating for about 5 min.

(5) Acquisition of infrared images

Adjust the focal length of the infrared camera lens until the concrete specimen appears clearly and steadily in the computer, then start the infrared image acquisition. In order to observe the obvious temperature changes on the surface of the concrete, this experiment is a total of 800 s of an infrared image video stream.

(6) Save and record the image data

Save the data in the form of video captured by the infrared camera to a designated location for subsequent image and data processing.

(7) Turn off the heat source

Turn off the infrared bulb switch and stop heating so that the temperature of the concrete specimen drops to the initial temperature.

(8) Concrete surface silane spraying

When the above-mentioned concrete specimen A_1_ surface temperature drops to the initial temperature, the concrete surface is sprayed with silane. The concrete specimen after spraying silane is recorded as A_2_.

After the surface of the concrete specimen is completely infiltrated with silane, the environmental conditions during the experiment are controlled so that specimen A_2_ and specimen A_1_ are tested under the same experimental environment. Repeat the above steps (1) to (7) to collect the infrared thermal image data of the concrete specimen A_2_ sprayed with silane and save it.

### 2.2. MATLAB-Based Infrared Image Processing

Infrared images characterize the infrared radiation distribution of an object. The overall grayscale distribution of the image is low, and the image is prone to grayscale concentration. In addition, during the actual imaging process, noise sources from the external environment may lead to multiple noises in the infrared image, which affects the information of the infrared image itself. Therefore, a series of image processing is needed for the IR images.

#### 2.2.1. Image Pre-Processing

Before image analysis, pre-processing operations need to be performed according to its features to enhance the contrast of the image. This facilitates the extraction of image features in the subsequent processing and improves the accuracy of the determination results. The pre-processing process generally includes image quality evaluation, image noise removal, image sharpening, etc. [38,39,40].

Commonly used objective evaluation metrics include entropy, standard deviation, etc. Among them, entropy reflects the information contained in the image [41]. The higher the entropy is, the more information the image contains. The standard deviation represents the dispersion of the grayscale of an image relative to the average grayscale of the image. The larger the standard deviation, the more hierarchical the image is. Based on the entropy and std2 function in MATLAB, the evaluation index values of the three frames with better subjective evaluation quality in the infrared images in the experimental group A are derived, respectively. In addition, we select the best infrared image quality for image analysis.

The noise of the image signal can be regarded as an interference signal, which has a random nature. Statistical features are commonly used to describe the noise, such as distribution functions and probability density functions of mean, variance, correlation functions, etc. In this study, the median filtering method is used to remove it [42]. The median filtering operation is implemented based on the medfilt2 function in MATLAB.

In order to extract the features of the concrete surface area, enhance the contour shape of the image, and make the edge information of the concrete image more prominent, the image sharpening method is used for the corresponding processing. The gradient method is often used in MATLAB to achieve image sharpening [43]. For an image *f*(*x*, *y*), the gradient at (*x*, *y*) is defined as
(1)$$grad(x,y)=(f^{\prime }{}_{x},f^{\prime }{}_{y})=\left(\frac{\partial f(x,y)}{\partial x},\frac{\partial f(x,y)}{\partial y}\right),$$
where the gradient is a vector quantity with direction and magnitude. There are several different computational operators depending on how the gradient size is calculated. The common ones are mainly the Laplacian operator, Prewitt operator, Sobel operator, etc. Among them, the Sobel operator adds the factor of weighted average to make the pixels on both sides of the edges in the image smooth. The obtained pre-processing images are shown in Figure 3.

By pre-processing the infrared image, the concrete image edge information is clearer and the surface area detail features are more obvious. A good foundation was laid for the subsequent edge detection processing.

#### 2.2.2. Concrete Surface Feature Recognition

In image processing, the region of interest in an image is generally referred to as the target (ROI), while the rest of the image is referred to as the background. Based on this principle, edge detection and segmentation of image targets can be achieved. In infrared images, the grayscale value of pixels at the target edge changes significantly, creating a discontinuity. For edges with step changes, the first-order derivative can be used to detect whether a point in the image is an edge point. The second-order derivative can be used to determine whether a pixel point lies to the left or right of an edge.

In image processing, edge detection can be accomplished by convolution with differential operators [44,45]. Differential operators are divided into first-order differential operators and second-order operators. Commonly used first-order operators include the Roberts operator, Prewitt operator, and Sobel operator. Commonly used second-order operators include the Log operator, Laplacian operator, and so on. In this study, the second-order Log operator was used for edge detection processing.

At present, a large number of analytical and computational methods have been formed for image segmentation, such as segmentation techniques by setting thresholds, segmentation based on genetic algorithms, etc. Among them, the image segmentation based on the threshold method has the advantages of large applicability, high operational efficiency, and speed [46]. The calculation formula of the threshold method is:(2)$$g(i,j)=\{\begin{array}{cc} 1 & f(i,j)\geq T\\ 0 & f(i,j)\leq T \end{array},$$
where *g*(*i*, *j*) denotes the pixel value of the input image; *f*(*i, j*) denotes the pixel value of the output image; and *T* represents the threshold.

When for the target pixel of the image *g*(*i*, *j*) = 0, the background pixel of the image *g*(*i*, *j*) = 1. When the threshold value is determined, the grayscale value of each pixel point in the image is compared with the threshold value. In turn, two different regions of the image segmentation can be obtained.

An iterative algorithm is used for the calculation of the threshold values. The maximum gray value and the minimum gray value of the image were obtained based on the gray statistics of the concrete infrared image. In addition, the threshold value *T* = 0.5373 was obtained. After determining the threshold size, the image segmented by the iterative thresholding method was obtained based on MATLAB.

#### 2.2.3. Morphological Processing of Images

There are incomplete edges and voids in regions present in the threshold-segmented infrared images of concrete. The binary image is processed by the morphological method. The processing operation of binary images can be realized by using the mathematical morphology method. Based on the closed operation in image morphology processing, MATLAB is applied to connect the broken edges and fill the holes, etc. [47]. The morphologically processed concrete image is obtained, as shown in Figure 4.

### 2.3. Evaluation of Uniformity Based on Cluster Analysis

#### 2.3.1. Clustering Analysis Algorithm

Combining the principles of algorithm selection and the characteristics of each algorithm, this study adopts the k-means algorithm for clustering analysis of pixel point temperature data. The k-means algorithm firstly selects *k* initial centroids randomly as clustering centers in a determined object containing n samples. The Euclidean distance is used to calculate the distance between other samples and the cluster center, and the sample points are grouped into the class with the closest distance to the cluster center. Then, the mean value of each class is used as the new clustering center, and the samples are re-classified into *k* classes [48,49]. Thus, iterative calculations are performed until the cluster centroids no longer change. The *k* value is the number of classes, which is determined by the sum of the squared errors (*SSE*) [50].
(3)$$SSE=\sum_{i=1}^{k}\sum_{p\subseteq C_{i}}{\left|p-m_{i}\right|}^{2},$$
where $C_{i}$ denotes the *i*-th cluster, *p* is any sample point in $C_{i}$, and $m_{i}$ denotes the mean point of all samples in $\;C_{i}$. *SSE* is the sum of squared clustering errors for all samples. The degree of merit of the cluster analysis results can be measured, and thus the appropriate *k* value can be determined.

#### 2.3.2. Calculation of Pixel Point Affiliation Based on Bayes Discriminant

In order to determine the loss of misclassified concrete surface pixel points, the posterior probability that a pixel point belongs to some overall, i.e., the affiliation degree, is obtained. The Bayes discriminant method is used for discriminant analysis of the classification of each pixel point on the concrete surface [51]. The basic idea is to assume the prior probability distribution of a known object. Then, the hypothesis is corrected by some actual sample of the known object. Finally, the posterior probability distribution is obtained.

Suppose *k* p-dimensional overall $G_{1},G_{2},\ldots ,G_{k}$, with density function $f_{j}(x)$, and prior probability $p_{j}=P(G_{j})$, *j* = 1, 2, 3, …, *k*. Note that the posterior probability function of the sample x judged to belong to $G_{i}$ is [52]:(4)$$P\left(k|X\right)=\frac{q_{k}f_{k}\left(X\right)}{\sum_{i=1}^{g}q_{i}f_{i}\left(X\right)}\left(k=1,2,\ldots ,g\right),$$
where $q_{i}$ is the prior probability of the sample *i* overall. A series of mathematical transformations of the above function yields the Bayes discriminant function as:(5)$$F_{k}\left(X\right)=X^{T}C_{k}+C_{ok}+\ln\;q_{k}=\sum_{j=1}^{m}C_{jk}x_{j}+C_{ok}+\ln\;q_{k},$$

The posterior probability that sample *x* belongs to *k* is the largest when the maximum $F_{k}\left(X\right)$ is taken, and then sample *x* is determined to belong to the *k*th overall.

#### 2.3.3. Evaluation of Homogeneity of Silane Coating

In order to reduce the influence of the concrete specimen’s own characteristics on the experimental results, the temperature pixel points of the specimen surface before and after spraying silane were clustered and discriminated separately. It is assumed that the clustering values of all pixel points on the concrete surface before spraying silane are set Q, and after spraying silane are set D. The absolute value of the difference between the two is set Z = |Q − D|. If the concrete surface is uniformly coated with silane, the clustering results of the same pixel on the concrete surface before and after the spraying of silane should be the same, i.e., Z = 0. Therefore, the uniformity of silane spraying on the concrete surface can be judged qualitatively by counting the difference in the clustering results of each pixel.

Due to the different affiliation of each pixel point obtained from Bayes discriminant calculation, the clustering difference values of each pixel point on the concrete surface before and after spraying the coating are also different. In order to calculate the coating uniformity more accurately, the weight value $\omega_{i}$ of the clustering difference of each pixel point is first calculated using hierarchical analysis [53,54]. Then, the uniformity of the concrete surface coating is calculated according to the magnitude of the weights, and the uniformity level can be determined. The formula for calculating the non-uniformity of the concrete surface coating can be derived as:(6)$$U=w_{1}\frac{\sum_{j=1}^{n_{1}}\varepsilon_{j}}{M}+w_{2}\frac{\sum_{j=1}^{n_{2}}\varepsilon_{j}}{M}+\cdots +w_{i}\frac{\sum_{j=1}^{n_{i}}\varepsilon_{j}}{M},$$
where *U* denotes the unevenness of the concrete surface coating; *n_i_* is the total number of pixel points at Z = *i*; *ω*_1_, *ω*_2_, …, *ω_i_* are the corresponding weight values; $\varepsilon_{j}$ denotes the affiliation degree corresponding to the *j*th pixel point; *M* denotes the total number of pixel points on the concrete surface.

In order to describe the uniformity grade situation of the concrete surface coating, the concrete surface coating uniformity grade can be divided into four grades according to the calculated inhomogeneity *U*. The specific division grades are shown in Table 1.

## 3. Results

### 3.1. Infrared Imaging Test Results

A total of 800 s of infrared image video streams were acquired for each experiment. Then, the IR image processing software ALTAIR was used to export the IR thermal images of the temperature distribution on the surface of the test block. The results of the initial state t = 0 and the final state t = 800 s of the experimental group N1 are shown in Figure 5 and Figure 6 as an example.

In order to facilitate the observation and comparison of the silane on the concrete surface temperature change pattern during the heating process, the graph of the average temperature of the concrete surface with the number of frames is drawn, as shown in Figure 7.

By comparing the surface average temperature change law of concrete before and after spraying silane. It can be seen that the average temperature rise rate of concrete structure is accelerated after spraying silane protective coating. This indicates that when silane coating forms a coating structure on the concrete surface, it affects the thermodynamic parameters such as specific heat capacity and thermal conductivity of the concrete structure itself. In addition, when the amount of sprayed silane varies, the degree of its influence will also vary.

### 3.2. Infrared Image Processing Results

To facilitate the extraction of features from concrete infrared images, different regions of the images need to be labeled. This facilitates the extraction and calculation of the image features. Since the obtained concrete images are binarized, the threshold value can be set to 0. In order to obtain the effect of the labeling process, the labels are highlighted in MATLAB in different colors to form a pseudo-color image. The image after processing is shown in Figure 8.

The area S of each area in the image is used as the basic feature of the infrared image. It is also used as an analytical index to determine the degree of uniformity of the concrete surface coating. In the above Figure 8, 1 and 2 represent different areas of the concrete surface, and the pixel value size of all pixels in area 1 is 1, while the pixel value size of all pixels in area 2 is 2. Therefore, the area size of each area can be calculated by counting the number of pixels in the image with pixel value size 1 and 2, respectively. In turn, the uniformity of the concrete surface coating can be analyzed qualitatively.

### 3.3. Cluster Analysis Results of Pixel Point Temperature Index

Taking the experimental group N1 as an example, the infrared images of concrete with uncoated silane on the surface are intercepted. The temperature data at different moments corresponding to the corresponding area are derived separately. The region includes a total of 17,098 pixel points. The temperature values corresponding to the moments of the 5000th, 10,000th, 15,000th, and 20,000th frames are denoted by *y_1_*, *y_2_*, *y_3_*, and *y_4_*, respectively, as the indicator variables for each sample point. The final results of the pixel point temperature data statistics are obtained and shown in Table 2.

The clustering analysis for the pixel point temperature data was implemented using the k-means algorithm. The *k*-values are taken as 2, 3, 4, 5, 6, 7, 8, 9, and 10 in order. Moreover, the distance of each pixel point to the cluster center in the cluster it belongs to is calculated by applying the SPSS Statistics 19 software. Then, the summation was obtained to obtain the magnitude of the error sum of squares corresponding to each different value of *k*, and the related line graphs were drawn, as shown in Figure 9.

From the above Figure 9, as the value of the number of clusters *k* increases, the finer the temperature samples are divided. Therefore, the cohesiveness of the data in each cluster will gradually increase, and the error squared and SSE will naturally become smaller. *k* = 5 is the optimal value for the cluster analysis of pixel point temperature data.

Once the *k*-values in the algorithm were determined, iterative and classification calculations were performed on the uncoated silane concrete surface temperature data. The four indicator temperature values of the final clustering center and the clustering results of each pixel point are obtained and shown in Table 3 and Table 4.

The clustering results show that each pixel point on the concrete surface is clustered into five categories according to the distribution of temperature. Categories I~V indicate the temperature in order from high to low. It can be understood that the concrete surface forms a high-temperature zone, a higher temperature zone, a medium temperature zone, a lower temperature zone, and a low-temperature zone. The number of pixel points in each of these classes is valid. The total number of pixel points in each class was counted, and the results are shown in Figure 10.

We can see from Figure 10 that the largest number of concrete surface pixel points are located in Class IV, accounting for about 32.88% of the entire concrete surface. This is followed by Class III and Class V, accounting for 28.37% and 26.61%, respectively. Class I has the least amount, accounting for only 2.52%. This shows that the medium on the uncoated concrete surface is affected by inhomogeneity, which leads to higher temperatures in some areas than in the general area. In addition, it shows that this clustering result is consistent with the temperature distribution in the infrared images.

### 3.4. Pixel Point Affiliation Calculation Results

Based on the clustering analysis results obtained earlier, Bayes discriminant analysis is calculated assuming that the prior probabilities of the points of each overall sample are equal. The formula for calculating the value of the function *T* for a given pixel point is obtained as follows:(7)$$\{\begin{array}{c} \begin{array}{l} T_{1}=-3044.930+190.483y_{1}-215.197y_{2}-401.228y_{3}+606.562y_{4}\\ T_{2}=-2597.086+178.761y_{1}-205.079y_{2}-403.184y_{3}+595.286y_{4} \end{array}\\ T_{3}=-2390.325+178.950y_{1}-216.994y_{2}-393.713y_{3}+589.982y_{4}^{}\\ T_{4}=-2252.997+179.037y_{1}-220.916y_{2}-390.832y_{3}+585.895y_{4}\\ T_{5}=-2096.145+178.281y_{1}-224.798y_{2}-387.047y_{3}+580.708y_{4} \end{array},$$

The *y*_1_, *y*_2_, *y*_3_, and *y*_4_ values of the unknown pixel points are brought into the above equation. The values of *T*_1_, *T*_2_, *T*_3_, *T*_4_, and *T*_5_ were calculated. Moreover, we compared the magnitude of these five values and took the category corresponding to the maximum value as the Bayesian discriminant result of the pixel point. The values were brought into the posterior probability calculation formula to obtain the posterior probability of each pixel point being discriminated into each category. To determine the cluster affiliation of any pixel point on the concrete surface, the maximum posterior probability value of the pixel point was taken as the affiliation of the pixel point belonging to the corresponding category. Figure 11 represents the two-dimensional distribution of clustering results for each pixel point.

In Figure 11 above, each different color represents a different overall class of pixel points clustered into. The individual dots represent the point coordinates of each pixel point projected from the 4-D coordinate system to the 2-D coordinate system. The group center of mass represents the coordinates of the mean projection of the pixel points in each aggregate to the two-dimensional plane. It can be seen that the distance from the sample points in each aggregate to the respective aggregate center of mass is the smallest. This also further verifies the accuracy of the clustering results obtained from the discriminant analysis. It is not difficult to find that the separation of individual pixel points is larger in the high-temperature region and the higher temperature region. This indicates that there are small areas of the concrete surface that are partially defective, leading to discontinuities in the concrete surface temperature and causing a sudden temperature rise.

### 3.5. Coating Uniformity Analysis

The clustering results and the affiliation degree of each pixel point on the concrete surface before and after the coating were calculated for the specimens of the above experimental group N1. The results are shown in Table 5, where the affiliation is the average of the maximum posterior probability of each pixel point.

The distribution of the variable Z is obtained from the results of the pixel point clustering statistics. The distribution characteristics roughly reflect the uniformity of the surface silane spraying to some extent. The total number of pixels is 6147, 3440, 2179, and 1345 when Z = 1, 2, 3, and 4. Finally, the unevenness of the surface coating of concrete specimens in experimental group N1 can be calculated as *U* = 0.122. The uniformity of the concrete surface coating in this group is evaluated as more uniform. Then, the inhomogeneity of the surface coating of concrete specimens in experimental group N2 and experimental group N3 are calculated, respectively. The results of their evaluation are obtained and shown in Table 6.

It can be seen from Table 6 that the evaluation results obtained from the evaluation method of concrete surface coating uniformity based on the combination of cluster analysis and hierarchical analysis are more consistent with the actual spraying situation. Therefore, the method is reasonable and feasible for evaluating the uniformity level of concrete coating. This can provide some reference value for the uniformity testing and judging in the actual project.

## 4. Discussion

This study conducted research on concrete surface coating uniformity detection and its determination and evaluation method. A nondestructive inspection and evaluation method of coating uniformity based on infrared imaging is proposed. We analyzed the images obtained from the non-destructive testing of infrared imaging by using MATLAB software. The characteristics of the infrared images of the concrete surface were obtained, and a method for determining the uniformity of the concrete surface coating by infrared images was proposed. We extracted the temperature distribution data of concrete surface pixel points and established a new evaluation method of concrete surface coating uniformity based on a combination of cluster analysis and hierarchical analysis. The results show that the determination results obtained by the method are consistent with the actual situation.

We compared the two methods based on MATLAB infrared image analysis processing and based on cluster analysis to determine the uniformity of the concrete surface coating. The results obtained from the image-processing-based method are more influenced by environmental factors. The method requires that the concrete structure itself is homogeneous, and also ensures that the concrete surface is heated completely uniformly. Therefore, the accuracy of the determination results obtained by this method is low. For the proposed new method of evaluating the uniformity of surface coating based on cluster analysis, the advantage of this method is that it can eliminate the influence of the characteristics of the concrete specimen itself on the evaluation results. The accuracy of the determination results is also higher.

At present, the application of infrared imaging non-destructive testing technology to detect the uniformity of concrete surface coatings is still in the preliminary stage. The factors affecting the infrared inspection results are many and complex. Coating uniformity testing experiments and processing methods for infrared images are not mature. Therefore, more in-depth research is needed to detect the uniformity of concrete surface coating more scientifically, accurately, and quickly. The proposed evaluation method is based on a large sample of temperature data of concrete surface pixel points. When the number of samples is small, the evaluation results obtained by the method may have some errors. Therefore, a more in-depth study of the uniformity evaluation method can be conducted when the number of samples is small. In addition, the evaluation results obtained are a qualitative uniformity rating rather than a quantitatively accurate description of the results. The quantitative study for the uniformity of concrete coatings also needs to be further investigated.

## 5. Conclusions

In this study, the uniformity of concrete surface coating is taken as the research object. We obtained the infrared thermal images of the concrete surface by the experiment of infrared imaging inspection of the concrete structure. The results obtained from the experiments are processed by applying the statistical methods of MATLAB image processing and cluster analysis, respectively. Finally, we derived a method for determining the uniformity of the concrete surface coating. The main conclusions are as follows:

1. A new non-destructive testing method applicable to concrete surface coating uniformity detection is proposed, infrared imaging nondestructive testing technology.

2. The infrared images acquired in the experiment were analyzed and processed based on MATLAB software. We derived the final features of the concrete surface images.

3. A new method based on the combination of cluster analysis and hierarchical analysis is proposed to evaluate the study of the uniformity of concrete surface coating. The results show that the evaluation results obtained from the calculation are more consistent with the actual spraying situation.

In future research, the quantitative evaluation of concrete surface coating uniformity and the evaluation of temperature data for small samples of pixel points can be further investigated.

## Figures and Tables

**Figure 1 sensors-21-05652-f001:**
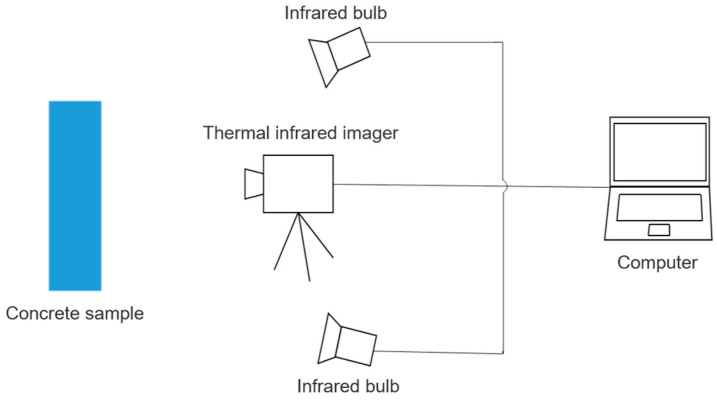
Infrared imaging detection system.

**Figure 2 sensors-21-05652-f002:**
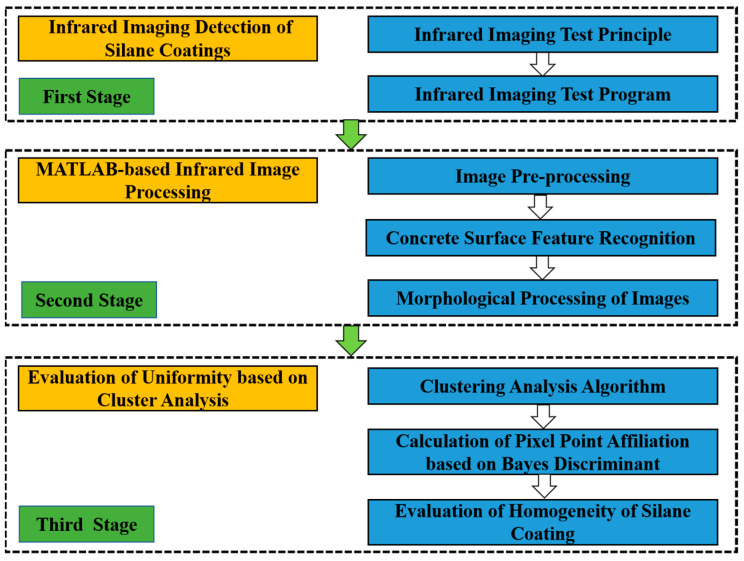
Flow chart of this study.

**Figure 3 sensors-21-05652-f003:**
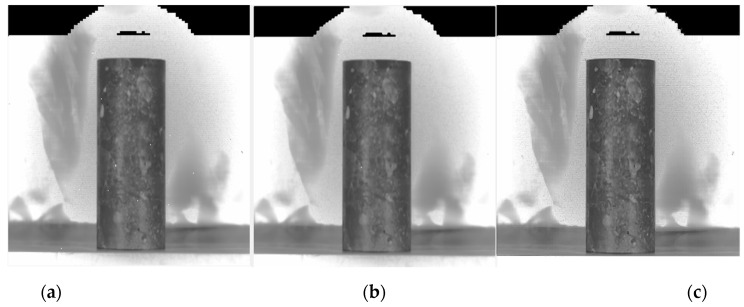
Concrete IR image pre-processing results: (**a**) image quality evaluation; (**b**) image noise removal; (**c**) image sharpening.

**Figure 4 sensors-21-05652-f004:**
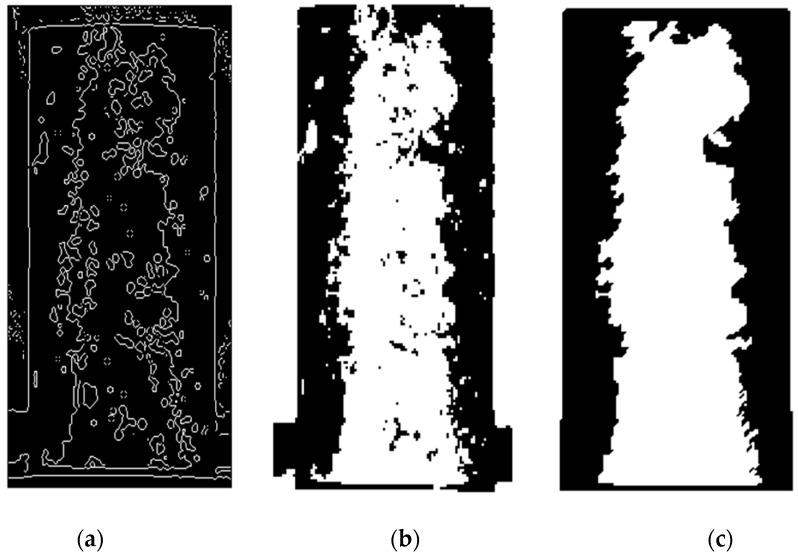
Concrete infrared image processing results: (**a**) edge detection processing; (**b**) binarization processing; (**c**) morphology processing.

**Figure 5 sensors-21-05652-f005:**
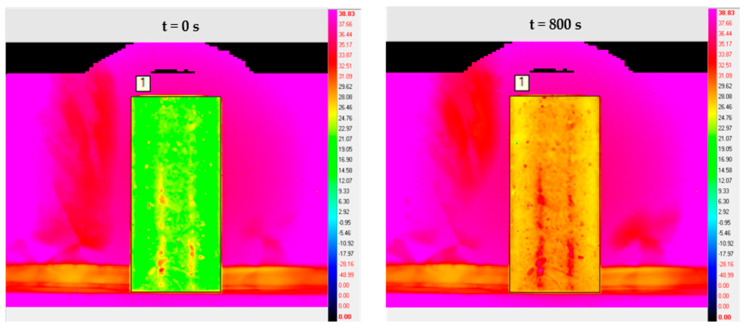
Infrared thermal image of the concrete surface before silane spraying.

**Figure 6 sensors-21-05652-f006:**
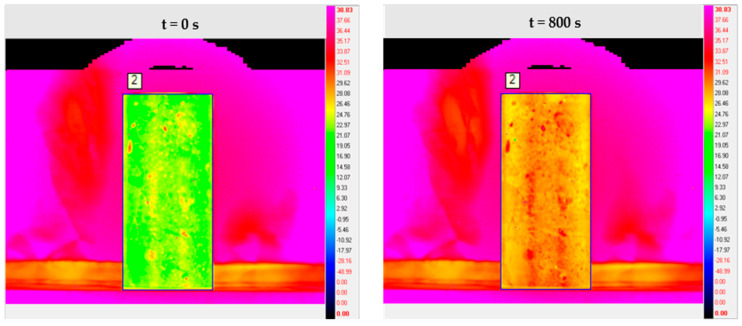
Infrared thermal image of the concrete surface after silane spraying.

**Figure 7 sensors-21-05652-f007:**
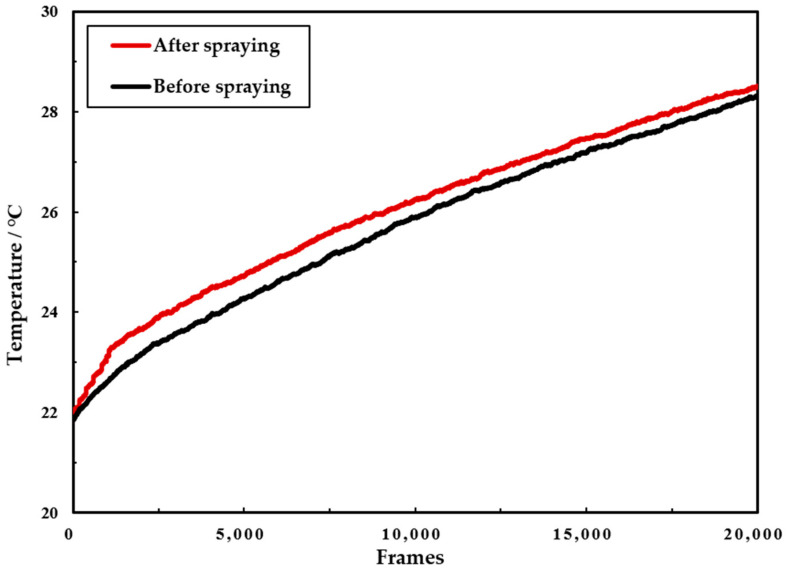
Average temperature–frame variation curve of concrete surface.

**Figure 8 sensors-21-05652-f008:**
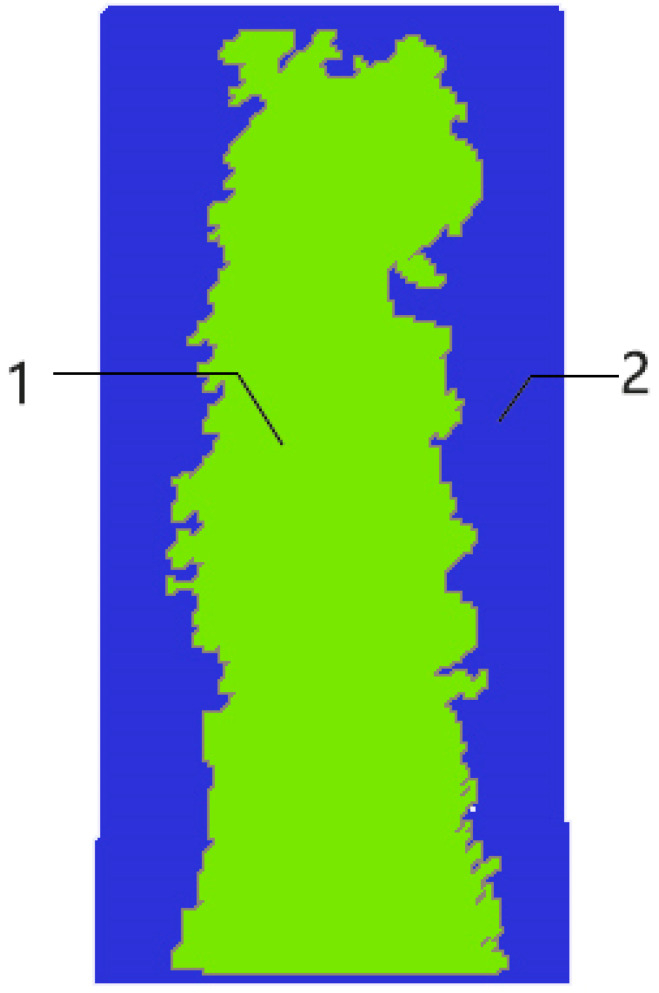
Pseudo-color labeled image.

**Figure 9 sensors-21-05652-f009:**
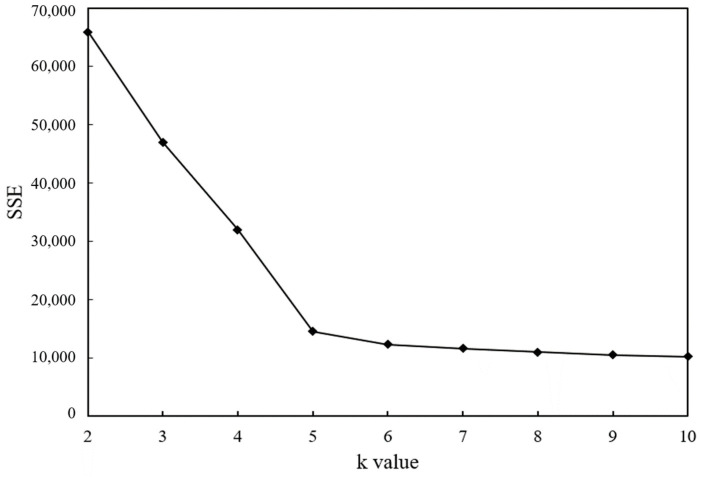
Relationship between SSE and *k* values.

**Figure 10 sensors-21-05652-f010:**
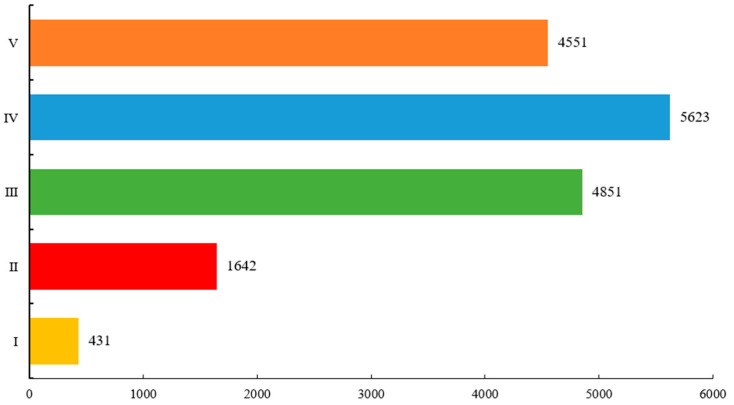
Number of pixel points in each cluster.

**Figure 11 sensors-21-05652-f011:**
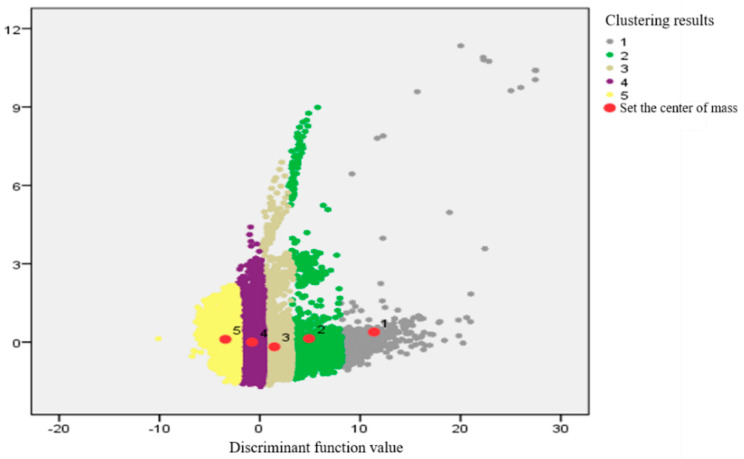
Cluster distribution of pixel points.

**Table 1 sensors-21-05652-t001:** Classification of concrete surface coating uniformity grade.

Uniformity Grade	Very Even	More Uniform	Uneven	Very Uneven
Unevenness *U*	≤0.1	0.1 < *U* ≤ 0.25	0.25 < *U* ≤ 0.4	>0.4

**Table 2 sensors-21-05652-t002:** Temperature statistics of each pixel point at different moments.

Sample Pixel Dots	*y*_1_/°C	*y*_2_/°C	*y*_3_/°C	*y*_4_/°C
1	34.01	35.22	35.27	35.29
2	30.76	32.11	32.44	32.64
3	28.47	29.85	30.41	30.85
4	26.72	28.14	28.87	29.48
5	25.46	26.93	27.80	28.51
6	24.60	26.08	27.05	27.81
7	23.75	25.26	26.29	27.15
8	23.01	24.53	25.65	26.57
9	22.87	24.36	25.55	26.45
10	22.43	23.99	25.25	26.16
⋮	⋮	⋮	⋮	⋮
17,098	27.97	28.35	29.11	29.56

**Table 3 sensors-21-05652-t003:** Final clustering center temperature values.

Indicators	Final Clustering Center				
	I	II	III	IV	V
*y*_1_/°C	29.64	26.58	24.93	23.92	22.70
*y*_2_/°C	31.11	28.15	26.56	25.54	24.33
*y*_3_/°C	32.27	29.41	27.92	26.93	25.75
*y*_4_/°C	33.11	30.35	28.93	27.97	26.82

**Table 4 sensors-21-05652-t004:** Clustering results of each pixel point.

Sample Pixel Dots	*y*_1_/°C	*y*_2_/°C	*y*_3_/°C	*y**_4_*/°C	Clustering Results
1	34.01	35.22	35.27	35.29	I
2	30.76	32.11	32.44	32.64	I
3	28.47	29.85	30.41	30.85	II
4	26.72	28.14	28.87	29.48	II
5	25.46	26.93	27.80	28.51	III
6	24.60	26.08	27.05	27.81	IV
7	23.75	25.26	26.29	27.15	IV
8	23.01	24.53	25.65	26.57	V
9	22.87	24.36	25.55	26.45	V
10	22.43	23.99	25.25	26.16	V
⋮	⋮	⋮	⋮	⋮	⋮
17,098	27.97	28.35	29.11	29.56	II

**Table 5 sensors-21-05652-t005:** Concrete surface pixel point clustering results before and after silane spraying.

Sample Pixel Dots	Pre-Painting Results	Results after Spraying	Affiliation
1	I	II	1
2	I	I	1
3	II	I	0.99985
4	II	IV	0.99830
5	III	V	0.87697
6	IV	V	0.80302
7	IV	V	0.62214
8	V	V	0.96985
9	V	V	0.98881
10	V	V	0.99867
⋮	⋮	⋮	⋮
17,098	II	II	0.99431

**Table 6 sensors-21-05652-t006:** Evaluation results of concrete coating uniformity in each experimental group.

Experimental Group	N1	N2	N3
Unevenness *U*	0.122	0.298	0.415
Homogeneity evaluation grade	More uniform	Uneven	Very uneven
Actual painting situation	More uniform	Uneven	Very uneven

## Data Availability

The data that support the findings of this study are available upon request from the authors.

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
