# Peer review of "Evaluation of the Uniformity of Protective Coatings on Concrete Structure Surfaces Based on Cluster Analysis"

_sensors, 2021, doi:10.3390/s21165652_

Round 1

Reviewer 1 Report

The paper entitles "The common ones are mainly Laplacian operator, Prewitt operator, Sobel operator, etc." addresses a nondestructive method based on infrared imaging and classic image processing combined with unsupervised learning to determine the uniformity of silane coating to protect the concrete surfaces from corrosion. The paper presents an interesting method that can be used in practice. However, the paper has a substantial lack of experimental details. The following is a list of the suggested information to be added to the paper:

  1. The experimental setup is incomplete. For example, there is no information about how the samples are made, the sample's health in terms of corrosion, the size of the samples, the contents of the sample, the distance between the sample and the camera, and more information like this type.
  2. How did the authors take into account the temperature variation on the surface? What did the authors do to ensure that the investigated area has a uniform heat flux and the temperature difference is merely due to the thickness of the protective coat?
  3. How can the heat flux to the concrete remain controlled in practice?
  4. What are the settings of the infrared imaging in terms of the emissivity of the surface and how it was adopted? How the emissivity of the samples changed after spraying more layers of the protective coating?
  5. The paper lacks quantitative discussions. For example, the thickness of the protective coating was classified as "uneven" or "even," this is not sufficient, knowing that a large computation was performed to get these descriptive conclusions. Maybe the thickness of the protective coating should have been calculated, and a profile of the coating thickness should have been provided. Maybe a thick enough uneven coating protects the member more effectively than a thin, even coating.
  6. Please add more references to the reader for the k-means and image processing methods.

Author Response

尊敬的女士/先生:

Thank you for your review of my paper. The following are my answers and corrections to the questions you have raised:

Point 1: The experimental setup is incomplete. For example, there is no information about how the samples are made, the sample's health in terms of corrosion, the size of the samples, the contents of the sample, the distance between the sample and the camera, and more information like this type.

Response 1: Thanks for your suggestions. We have supplemented the experimental design and procedure in Section 2.1.2 of revised paper. These include the origin and manufacture of the samples, the health of the samples, the size of the samples, the distance between the sample and the camera, and much more. 

Point 2: How did the authors take into account the temperature variation on the surface? What did the authors do to ensure that the investigated area has a uniform heat flux and the temperature difference is merely due to the thickness of the protective coat?

Response 2: If the protective coating is uniform on the surface of the sample, the density, thermal conductivity and specific heat capacity of the coating structure should change identically. The distribution of the temperature variation characteristics caused by the coating on the surface is also consistent. If the protective coating is not uniform, the distribution of temperature variation characteristics will be different. 

In the supplementary test, in order to ensure uniform heating on the sample surface, we adopt the uniform temperature field formed by two infrared bulbs, and preheat the infrared bulb and the thermal imager. In addition, we control that the surface environment of the specimen is consistent before and after silane spraying, so that the specimen is in the same experimental environment.

Point 3: How can the heat flux to the concrete remain controlled in practice?

Response 3: In the test, we symmetrically set up two infrared bulbs, which apply the same and constant heat flux to the test specimen surface. Some descriptions of infrared bulbs are added in the revised paper.

Point 4: What are the settings of the infrared imaging in terms of the emissivity of the surface and how it was adopted? How the emissivity of the samples changed after spraying more layers of the protective coating?

Response 4: The emissivity of concrete surface is affected by the properties of concrete and environment factors. Therefore, it is difficult to obtain accurate surface emissivity. According to the test data and related studies, the emissivity of concrete surface is 0.9. After spraying silane on concrete surface, the surface emissivity increases slightly.

Point 5: The paper lacks quantitative discussions. For example, the thickness of the protective coating was classified as "uneven" or "even," this is not sufficient, knowing that a large computation was performed to get these descriptive conclusions. Maybe the thickness of the protective coating should have been calculated, and a profile of the coating thickness should have been provided. Maybe a thick enough uneven coating protects the member more effectively than a thin, even coating.

Response 5: The suggestions put forward by reviewer are very meaningful, which are also the direction we need to work on. There is no unified quantitative evaluation standard for the uniformity of silane coating. In fact, if we use infrared imaging, we can only get subjective qualitative conclusions. Therefore, in order to obtain better quantitative evaluation results, we adopt cluster analysis method to analyse pixel temperature. Finally, we get the results similar to the quantitative analysis. However, there are certain subjective factors in uniform and non-uniform classification, and we need to improve these studies.

It is also quite true that you pointed out that thicker uneven coating protection may be better than thin uniform protection. But this will cause material waste and raise the cost of construction. This study is only one method for the uniformity evaluation of concrete silane coating, and it is the beginning of this field. I believe we will achieve certain results in quantitative evaluation.

Point 6: Please add more references to the reader for the k-means and image processing methods.

Response 6: For the k-means algorithm and image processing methods, we add seven references for readers.

Reviewer 2 Report

The manuscript is within the scope of the Sensors journal. However, before publication some major concerns must be addressed, namely:

  1. The introduction needs to be updated since it does not reflect the actual state of the art. Moreover the references are not well indicated.
  2. It is not clear how the experimental results support the simmulating results.
  3. Tables 2, 3 and 4 - Please indicate the errors
  4. Please turn the conclusions more shortned.

Author Response

Dear Ms/Mr,

Thank you for your review of my paper. The following are my answers and corrections to the questions you have raised:

Point 1: The introduction needs to be updated since it does not reflect the actual state of the art. Moreover the references are not well indicated.

Response 1: Thank you for your review. Your suggestion has a very positive effect on the improvement of the quality of the article. In the introduction, we improve the application of infrared thermal imaging technology in concrete detection. In addition, we adjuste the reference format according to the journal format requirements.

Point 2: It is not clear how the experimental results support the simmulating results.

Response 2: Based on the experimental results, we have got the pixel temperature of concrete surface before and after silane spraying. In the simulation process, we conduct clustering and discriminant analysis of temperature pixels before and after silane spraying at section 3.3 and 3.4 to qualitatively judge the uniformity of silane spraying on concrete surface.

Point 3: Tables 2, 3 and 4 - Please indicate the errors.

Response 3: In Tab. 2, 3 and 4, we have adjusted the units, significant numbers, and related formats of the data.

Point 4: Please turn the conclusions more shortned.

Response 4: Considering your suggestion, we have reduced the conclusion to less than 200 words.

Reviewer 3 Report

The purpose of the paper entitled Evaluation of the uniformity of protective coatings on concrete structure surfaces based on cluster analysis, is an interesting one. All chapters are clearly presented and well written although, for me, it is not clear if the methodology will always need to measure first the uncoated concrete, in order to obtain good results.

Some small observations are also presented as follows:

- line 12: ”measure” instead of ”measures”;

- the bibliographic references in text should be placed in square brackets: eg. [1];

- line 37: please rephrase;

- line 74-75: please rephrase;

- line 83: where is ”at home”?

- line 145: please use subscript for numbers in the chemical formula;

- line 145-146: please rephrase;

- line 180: please rephrase;

- line 255-256: please rephrase;

- line 378: please rephrase;

- lines 493-494: please rephrase.

Author Response

尊敬的女士/先生:

Thank you for your review of my paper. The following are my answers and corrections to the questions you have raised:

Point 1: It is not clear if the methodology will always need to measure first the uncoated concrete, in order to obtain good results.

Response 1: Thank you for your review. Because the characteristics of concrete specimens have great influence on the experimental results. We conducted clustering and discriminant analysis on the temperature pixels of the specimen surface before and after silane spraying. In this way, it is more accurate to judge the uniformity of silane coating on concrete surface. In practical application, it is not necessary to test the initial value before spraying for the same concrete, we can take the average value of multiple tests as the initial value.

Point 2: Some small observations are also presented as follows:

- line 12: ”measure” instead of ”measures”;

- the bibliographic references in text should be placed in square brackets: eg. [1];

- line 37: please rephrase;

- line 74-75: please rephrase;

- line 83: where is ”at home”?

- line 145: please use subscript for numbers in the chemical formula;

- line 145-146: please rephrase;

- line 180: please rephrase;

- line 255-256: please rephrase;

- line 378: please rephrase;

- lines 493-494: please rephrase

Response 2: We are very sorry for our incorrect writing. According to your suggestion, we have carefully modified the words, sentences and reference format of the article to make the article more rigorous.

Round 2

Reviewer 1 Report

The paper has been significantly improved, and the methods are described very well. The reviewer believes that the paper has a high quality as a peer-reviewer journal paper; hence, it is recommended for publication in "Sensors".

Reviewer 2 Report

All the issues raised were correctly addressed. Therefore, I support publication in the present form.